# Early-Life Fecal Transplantation from High Muscle Yield Rainbow Trout to Low Muscle Yield Recipients Accelerates Somatic Growth through Respiratory and Mitochondrial Efficiency Modulation

**DOI:** 10.3390/microorganisms12020261

**Published:** 2024-01-26

**Authors:** Guglielmo Raymo, Ali Ali, Ridwan O. Ahmed, Mohamed Salem

**Affiliations:** Department of Animal and Avian Sciences, University of Maryland, College Park, MD 20742-231, USA; graymo@umd.edu (G.R.); areali@umd.edu (A.A.); rahmed20@umd.edu (R.O.A.)

**Keywords:** aquaculture, gut microbe function, microbiota, selective breeding, fillet, rainbow trout, ARS-FY-H, ARS-FY-L

## Abstract

Previous studies conducted in our lab revealed microbial assemblages to vary significantly between high (ARS-FY-H) and low fillet yield (ARS-FY-L) genetic lines in adult rainbow trout. We hypothesized that a high ARS-FY-H donor microbiome can accelerate somatic growth in microbiome-depleted rainbow trout larvae of the ARS-FY-L line. Germ-depleted larvae of low ARS-FY-L line trout reared in sterile environments were exposed to high- or low-fillet yield-derived microbiomes starting at first feeding for 27 weeks. Despite weight-normalized diets, somatic mass was significantly increased in larvae receiving high fillet yield microbiome cocktails at 27 weeks post-hatch. RNA-seq from fish tails reveals enrichment in NADH dehydrogenase activity, oxygen carrier, hemoglobin complex, gas transport, and respiratory pathways in high fillet yield recolonized larvae. Transcriptome interrogation suggests a relationship between electron transport chain inputs and body weight assimilation, mediated by the gut microbiome. These findings suggest that microbiome payload originating from high fillet yield adult donors primarily accelerates juvenile somatic mass assimilation through respiratory and mitochondrial input modulation. Further microbiome studies are warranted to assess how increasing beneficial microbial taxa could be a basis for formulating appropriate pre-, pro-, or post-biotics in the form of feed additives and lead to fecal transplantation protocols for accelerated feed conversion and fillet yield in aquaculture.

## 1. Introduction

Host species selectivity for unique microbial intestinal profiles is understood in fish, and preference for species-specific microbiome assemblies is suggested to be independent of externalities such as diet and environment. The core microbiome of zebrafish raised in captive conditions has been demonstrated to mimic the intestinal fauna of wild zebrafish closely, indicating a host-mediated selective pressure on intestinal colonizers [1]. Moreover, a core microbiome has been characterized at 16s resolution in rainbow trout, common carp, Atlantic cod, and European sea bass [2,3,4,5]. Notably, the intestinal microbiomes of rainbow trout reared in geographically isolated and distinct farming facilities reflect strong similarities among unique populations despite varying influent water quality, bioburden, and feed [4].

Core microbiome composition is strongly conserved within host species; however, the abundance of individual bacterial operational taxonomic units (OTUs) has been shown to vary between genetic lines [6]. In tilapia, domestication for cold tolerance has been linked with specific microbial gut profiles [7]. In rainbow trout, short-chain fatty acid (SCFA) producing taxa such as *Bacteroidetes*, *Fusobacteria*, and *Deniococcus* may be associated with higher muscle accumulation as a function of their ability to convert indigestible polysaccharides to utilizable energy [8].

Despite survey sequencing efforts, comprehensive characterization of host-microbiome interplay in aquaculture species lags terrestrial livestock. Prebiotic feed formulations containing *Lactobacillus*, *Pediococcus*, *Bifidobacterium*, and *Enterococcus* have been thoroughly investigated in swine [9], poultry [10], and ruminant species [11,12]. Deleterious effects on growth and muscle development in microbiome-depleted murine models are well characterized. Germ-depleted mice demonstrated significant muscle atrophy concordant with reduced mitochondrial function and electron transport chain gene transcription [13]. Moreover, colonocytes in germ-free mice displayed decreased NAD/NADH+ concentrations, oxidative phosphorylation, TCA activity, as well as skeletal muscle autophagy [14]. Notably, a reversion to homeostatic transcription levels was observed for differentially expressed genes when exogenous butyrate was administered to germ-free mice, implicating native murine microbiota as an indispensable energy harvesting pathway. Despite work in murine model species, the direct role of gut microbiome composition on energy regulation and skeletal muscle recruitment remains unclear in agriculturally relevant animals, including rainbow trout.

Adult trout derived from high muscle yield genetic lines exhibited greater alpha diversity and heightened abundance of *Bacteroidetes*, *Fusobacteria*, and *Deniococcus,* which may play a role in SCFA production [6]. Nonetheless, the underlying directional causality between muscle mass accretion and enteric microbiome carbohydrate fermentation is not entirely understood in rainbow trout. In this experiment, we reared microbiome-depleted rainbow trout eggs in sterile conditions; we then administered fecal samples derived from either high- or low-muscle yield genetic line adult donors to larvae at the first feeding. We monitored growth over 27 weeks and subsequently conducted RNA-seq transcriptome profiling on whole tail sections of recolonized juveniles in tandem with dual 16s ribosomal amplicon sequencing of adult donor fecal samples and juvenile intestines to assess germ colonization efficacy.

Aquaculture is lauded as the fastest-growing food production sector, surpassing wild-caught fisheries by gross tonnage in 2014 [15]. Maximizing aquaculture production efficiency through conventional selective breeding allows for improved edible fillet yield and accelerated growth in rigorous farming contexts [16]. Current muscle phenotyping strategies in finfish present major practical drawbacks and animal welfare concerns, as fish are sacrificed to assess breeding value related to skeletal muscle composition and fillet quality traits. Non-invasive microbial gut sampling may assist in future precision breeding efforts as rainbow trout muscle yield genetic lines are shown to harbor distinct gut microbial assemblages [6]. Additionally, engineered pre-biotic supplementation may optimize feed utilization and animal health, minimizing industrywide environmental impacts stemming from antibiotic misuse and overfeeding.

## 2. Methodology

### 2.1. Fecal Recolonization

Low muscle yield genetic line (ARS-FY-L) eggs from a single genetic family were obtained from the National Center for Cold and Cool Water Aquaculture (NCCCWA, Leetown, WV, USA). The NCCCWA began a growth-selected line of rainbow trout in 2002 and has developed five generations strictly for enhanced growth performance, followed by four generations of selective breeding for fillet yield. This study used donors from the 2020 year class (YC) and eggs from the 2022 year class [17]. In an abiotic environment, eggs were reared following a germ-free husbandry protocol [18] outlined in Figure 1. Briefly, eggs were incubated in filtered (0.22 µm), aerated, and autoclaved water at 15 °C until hatching under gentle agitation. Rearing water was dosed with penicillin G (10,000 U/mL), streptomycin (10 mg/mL), kanamycin sulfate (100 mg/mL), and amphotericin B solution (250 μg/mL). Water exchange was conducted daily prior to egg hatching. Eggs were disinfected in an iodine bath (10 ppm) for 10 min, then rinsed thrice in temperature-matched, autoclaved water. Once hatched, yolk sack larvae were moved to sterile containers without antibiotics with continued daily water exchanges Fecal microbiome samples collected from apparently healthy high (ARS-FY-H) and low (ARS-FY-L) muscle yield donors at approximately 26 months of age housed in a common recirculating aquaculture system (RAS) were mixed with feed in a 2 mL tube after 15 s of vortexing. Feed laden with active microbiome was provided to larvae at first feeding and daily for 27 weeks. Fecal samples were collected from adult donors weekly by manually stripping adult donors. Fecal samples were stored in a 15 mL PBS solution at 4 °C and 3 mL were incorporated with recipient feed daily. Raw subsamples not kept in PBS were kept at −80 °C for microbiome analysis. Juvenile fish were fed 15% body weight daily throughout the experiment, and biometrics were recorded weekly by generating individual wet-weight measurements. At 27 weeks post-hatch, fish were euthanized, and whole tail sections from 11 juvenile fish were used for RNA-seq analysis.

### 2.2. 16s Microbiome Sequencing and Bioinformatics

Donor fecal samples were pooled by genetic family from each of the genetic lines ARS-FY-H/ARS-FY-L for sequencing, cumulating in 18 unique family samples that underwent individual sequencing. DNA was extracted from both larvae intestinal and donor fecal samples using the ZymoBIOMICS^®^-96 MagBead DNA Kit (Zymo Research, Irvine, CA, USA). Library preparation was conducted using a Quick-16S Plus NGS Library Prep Kit coupled with a v3-v4 primer set (Zymo Research, Irvine, CA, USA). Negative controls during both DNA extraction and library preparation served to normalize for bioburden during wet-lab processing. Sequencing was conducted using an Illumina MiSeq with a 10% PhiX spike-in (Illumina, San Diego, CA, USA). Raw reads were assembled into amplicons using the Dada2 pipeline, and chimeric reads were removed as well [19]. Taxonomy composition was constructed with the Qiime Uclust package (v.1.9.1) [20]. Alpha and beta diversity metrics and PCoA were performed with Qiime [21] and differences between operational taxonomic groups (OTU) were computed using the Linear discriminant analysis Effect Size (LefSE) approach using default settings [22].

### 2.3. RNA Sequencing and Bioinformatics

RNA was extracted from whole-tail tissue samples using Tri-Zol reagent (Invitrogen, Carlsbad, CA, USA) and nucleic acid quantification was conducted using a Qubit 4 Fluorometer and RNA high sensitivity assay (Invitrogen, Waltham, MA, USA). RNA purity was assessed using Nanodrop™ ND-1000 560 spectrophotometer (Thermo Fisher Scientific, Waltham, MA, USA) and integrity was verified with agarose gel. Library preparation was performed using xGenTM RNA Library Prep Kit (IDT, Coralville, IA, USA). Sequencing was conducted using an Illumina Novaseq 6000 (Illumina, San Diego, CA, USA), producing 150 bp paired-end reads at a minimum sequencing depth of 40 M reads. Raw sequencing reads were trimmed using Trimmomatic (v0.39) [23] to remove adaptors, followed by quality control using FastQC (v0.11.5) [24] and MultiQC (v1.14) [25]. The quality score threshold for samples to be used for differential expression analysis was set to Q30.

Processed reads were mapped to the rainbow trout reference genome (NCBI Accension GCF_013265735.2) using the CLC genomics workbench (version 22.0, Qiagen, Hillden, Germany). Reads were mapped with a cost for a mismatch equals 2, insertion or deletion equals 3. Differential expression was performed using the CLC genomics workbench RNA-Seq analysis tool. Transcript and gene threshold for significance was delineated with FDR  ≤  0.05 and absolute fold change value  ≥ 2. Pathway enrichment analysis was conducted using ShinyGO [26] configured to output GO cellular components, biological processes, and molecular function pathways at FDR ≤ 0.05. Heatmaps for RNA-seq data were generated using the gplots Heatmap.2 package (v3.1.1) [27], and differentially expressed genes displayed in the heatmap were selected for their biological relevance.

## 3. Results and Discussion

### 3.1. High Muscle Yield Recolonized Larvae Displayed Enhanced Growth

High-muscle (ARS-FY-H) yield gnotobiotic larvae displayed significantly greater muscle mass compared to low (ARS-FY-L) muscle yield recolonized larvae starting at 5 months post-hatch, and this trend was observed across the final two sampling points (Figure 2). The standard growth rate (SGR) was 1.47% and 1.37% of final biomass in the high and low groups, respectively. Notably, high muscle yield recolonized larvae displayed heightened mortality during the preliminary phases of the growth trial compared to low muscle yield conspecifics. Mortality was observed to plateau in both groups at 5 months post-hatch, culminating in 84% mortality in the high and 36% in the low (Figure 2). Reciprocal fecal transplantation of the wood tiger moth (Arctia plantaginis) showed a doubled mortality rate in a fast-to-slow growing genotype transplantation compared to a slow-to-fast growing genotype transplantation [28]. The host’s genetic incompatibility may at least partially explain the higher mortality observed in high muscle yield recolonized larvae of our experiment.

### 3.2. Bacterial Taxa from Adult Donors and Recolonized Recipient Larvae Intestine

Amplicon sequencing conducted on recolonized larvae intestines revealed a total of five phyla, 58 genera, 29 families, and 108 significantly differentially abundant bacterial species between the high- and low-muscle yield recolonized groups (Figure 3, Linear discriminant analysis Effect Size, *p* < 0.05, *LDA* > 2, Appendix A). A total of 30 OTUs were found to vary significantly between high and low muscle yield adult donors, of which 24 were more abundant in the high muscle yield group, and six in the low muscle yield group (Figure 3A, Appendix A).

Additionally, 82 species were more abundant in the high muscle yield recolonized group. The phylum *Chloroflexi* and *Actinobacteria* were more abundant in the high muscle yield recolonized group (Figure 3B). In contrast, the phylum *Firmicutes* and *Proteobacteria* were more abundant in the low muscle yield recolonized population, and the phylum *Fusobacteria* was absent in the intestines of high muscle yield recolonized trout.

Regarding the donor fecal samples, 16s amplicon sequencing conducted on 18 pooled donor fecal samples revealed more limited variability between muscle yield groups (Appendix A). A total of 30 OTUs were found to vary between muscle yield donor groups, of which 18 were mapped at species resolution. Notably, four shared OTUs were found between low muscle yield adult donors and juvenile recipients, and five shared OTUs were found between high muscle yield adult donors and juvenile recipients (Figure 3C). Genera *Leucobacter* and *Kocuria* of the phylum *Actinobacteria* were more abundant in both high muscle yield donors and juveniles than low muscle yield donors and juveniles (Figure 3C). In low muscle yield donors and recolonized larvae, Parachlamydia and Cetobacterium genera were variably abundant compared to high muscle yield groups (Figure 3C).

A Wilcoxon rank sum test was conducted on fisher alpha diversity, revealing significant compositional differences (*p* < 0.0001) between high and low muscle yield re-conventionalized larvae at 27 weeks of age (Figure 4). Notably, no significant differences in fisher alpha diversity were observed between the donor samples despite a shared trend for lower alpha diversity in both groups. Beta diversity generated for larvae revealed strong clustering along a principal component axis (PC1, 81.94%); however, no pronounced beta clustering was observed in donor intestines (Figure 4). These findings may indicate an exaggerated bacterial diversity in early life stages compared to donors, suggesting trout to be variably receptive to bacterial colonization depending on life stage.

A moderate relationship between muscle yield genetic line and body weight is reported in rainbow trout (R^2^ = 0.56) and the increased abundance of *Firmicutes* in the low (ARS-FY-L) muscle yield genetic line may explain the lower body weight observed in recolonized larvae [29]. *Leuconostoc citreum* pertaining to the phylum *Bacillota* was found to be more abundant in the high muscle yield recolonized group (*LDA* = 2.87, *p* = 0.036) and it is classified as highly probiotic due to pathogenetic suppression attributes [30]. Specifically, *L. citreum* expresses epithelial cell wall anchoring proteins promoting non-transient colonization of the host intestine, and in humans, *L. citreum* supplementation is shown to increase lipolysis [31]. *Staphylococcus equorum* was similarly found in higher abundance in the high muscle yield recolonized line (*LDA* = 3.098, *p* = 0.036), and is known to produce free fatty acids through proteolytic and lipolytic degradation pathways [32]. Increased abundance of both *L. citreum* and *S. equorum* in the high muscle yield population may explain, in part, the notable weight difference observed between experimental groups.

*Clostridium botulinum* was found to be more abundant in the high muscle yield recolonized larvae group (*LDA* = 2.75, *p* = 0.036, Appendix A) as well as *Clostridium Botulinum Haemolyticum* (*LDA* = 2.626, *p* = 0.035, Appendix A). The threefold increase in mortality observed in the high muscle yield recolonized larvae group may be explained by the abundance of *Botulinum* spp. in this population, which has been linked to mortality in a wide range of freshwater teleost species [33]. Moreover, *Botulinum* spp. was not detected in the low muscle yield (ARS-FY-L) group. Mortality in the high (ARS-FY-H) muscle yield recolonized group was most pronounced during the first 13 weeks of the experiment. Early-life stage rainbow trout are known to be more susceptible to bacterial infections, with limited immune response ability via dynamic gene regulation and transcription modulation compared to mature conspecifics [34]. For instance, enteric red mouth disease (ERM) induced by *Yersinia ruckeri* has been demonstrated to be highly lethal in juvenile rainbow trout despite susceptibility across life stages [35]. Future efforts to improve rainbow trout growth and muscle accretion via fecal transplantation or microbiome manipulations may require additional screening and filtering of samples containing harmful bacteria such as *Clostridium botulinum*.

*Cetobacterium somerae* was found to be significantly abundant in both low muscle yield recolonized larvae and upstream adult donors (Appendix A). Notably, *C. somerae* is recognized for its ability to produce vitamin B_12_ [36]. In vivo supplementation of *C. somerae-derived* B_12_ results in demonstrated alterations to whole gut redox status, reducing the oxygen concentration in the intestinal environment. Oxygen-saturated environments have been shown to promote pathogenic bacterial virulence [37]. Non-fluctuation intestinal redox status impacts gut microbiome compositional stability and functional profile resiliency [38]. Additionally, B_12_ has been shown to safeguard against pathogenic infection, specifically promoting gut barrier tight junction activity in zebrafish by increasing the expression of canonic tight junction proteins (Claudin15, Occludin, and Zo-1) [38]. Pronounced mortality in the high-yield recolonized group observed after antibiotic treatment and microbiome cocktail administration may be explained, in part, by the absence of *C. somerae* and the negative implications of aerobically saturated intestinal environments.

Gut microbial profiling in adult donor rainbow trout revealed several notable SCFA-producing taxa to be differentially abundant in the high muscle yield line. *Clostridia lachnoclostridium* was found to be more abundant in fish belonging to the high muscle yield genetic line (*LDA* = 2.05, *p* = 0.025, Appendix A). *C*. *lachnoclostridium* is known to primarily hydrolyze complex, indigestible polysaccharides into butyrate [39]. Similarly, *Lactobacillus salivarius* was found in greater concentration in the high muscle yield donor lineage (*LDA* = 3.3, *p* = 0.037, Appendix A). Propionate and butyrate formation was shown to increase substantially in both simulated chicken cecum models and the SHIME (Simulator of Human Microbial Ecosystem) reactor [40,41]. Additionally, high muscle yield recolonized larvae exhibited upregulation of matrix metallopeptidase 9 (*mmp9*) integrin beta 3 (*itgb3b*) coagulation factor XIII A1 polypeptide (*f13a1b)* and adrenoceptor alpha 2B (*adra2b*), which are all implicated in gene ontology (GO) biological pathways for wound healing. Exogenous butyrate supplementation is demonstrated to improve response to wounding in zebrafish via the recruitment of neutrophils and pro-inflammatory macrophages [42]. We suspect additional butyrate supplemented by high muscle yield-specific gut microbes to confer added host resilience to mechanical damage and wounding. However, supplemental experimental data are required to quantify the bulk SCFA content produced in vivo and investigate if a notable difference is observed between transplantation groups.

*Lactobacillus ingluviei* was found to be more abundant in high muscle yield donors (*LDA* = 2.76, *p* = 0.039) and is reported to cause significant weight gain in germ-free chicks, ducks, and mice [43,44]. *L. ingluviei*-mediated weight gain is induced by hepatic steatosis [45]. Despite our oversight in considering liver triglyceride content or hepatic specific transcriptome response when assessing weight gain in recolonized larvae, whole tail RNA-seq (discussed below) revealed upregulation of GRB10 interacting GYF protein 2 (*GIGYF2*). *GIGYF2* is characterized as an endogenous negative regulator, acting in concert with growth factor receptor-bound protein 10 (*GRB10*), to inhibit insulin-like growth factor 1 receptor (*IGF1R*) transmembrane signaling [46]. *GIGYF2* upregulation and concurrent reduction in *IGF1R* may play a role in the etiology of hepatic steatosis and somatic weight gain. Notably, in skeletal muscle, the development of insulin resistance is associated with decreases in oxidative phosphorylation capacity, whereas we observed an inverse phenomenon [47].

In our experiment, unclassified *Bacteroidetes* were found to be more abundant in the high (ARS-FY-H) muscle yield recolonized population (*LDA* = 2.67, *p* = 0.039, Appendix A), whereas unclassified *Firmicutes* were found to be more abundant in the low (ARS-FY-L) muscle yield recolonized fish (*LDA* = 5.19, *p* = 0.006, Appendix A). Both of these findings concord with our previous work on the microbial composition of adult rainbow trout from high (ARS-FY-H) and low (ARS-FY-L) genetic lines [6]. The ratio of Firmicutes to Bacteroidetes plays an important role in SCFA synthesis and obesity in humans [48]. We found the ratio of Firmicutes to Bacteroidetes to vary significantly between the high (ARS-FY-H) and low (ARS-FY-L) muscle yield recolonized populations (*p* = 0.004861, *n* = 8, Appendix A) with the low muscle yield recolonized group reflecting a higher ratio of Firmicutes to Bacteroidetes.

### 3.3. Insight into Gene Expression of Recipient Larvae

Gene expression analysis was conducted on five fish in the high muscle yield recolonized larvae and six fish in the low muscle yield recolonized larvae. The whole tail sections, including epidermis, muscle, vasculature, and bone, were collected 27 weeks post-hatch for transcriptome interrogation. A total of 318 transcripts (FDR *p* < 0.05, Fold Change ≥ 2, Appendix A) were found to be upregulated when comparing high muscle yield larvae to low muscle yield larvae, and 529 transcripts were found to be downregulated (FDR *p* < 0.05, Fold Change ≤ −2, Appendix A). In total, 41 significantly enriched pathways (FDR *p* < 0.05, Figure 5, Appendix A) were reported using ShinyGO [26] when considering upregulated genes in the high muscle yield group as input terms. Notably, six of these pathways reflected significant enrichment in respirasome, oxygen transport, oxygen binding, and hemoglobin complex activity (Figure 5). Notable genes in the aforementioned enriched GO terms include hemoglobin alpha subunit D (*hbad*), and hemoglobin beta (*hbd1*), (Figure 6, Appendix A). It remains to be determined if accelerated oxygen transport and respirasome activity observed here are the underlying causative element responsible for weight gain or serve as proxy biomarkers for specific dynamic action (SDA) as oxygen intake is shown to share a linear relationship with feed intake in teleost [49].

Another major subset of pathways consisting of five significantly enriched GO terms implicated electron transport chain activity within the high muscle yield recolonized group. Specifically, oxidoreductase activity acting on NAD(P)H, oxidoreduction-driven active transmembrane transporter activity, NADH dehydrogenase quinone, and ubiquinone activity were all found to be enriched (Figure 5). Moreover, individual transcripts coding for NADH dehydrogenase subunits 1, 5, and 6 were all found to be highly upregulated in high muscle yield recolonized recipients, as well as NADH: Ubiquinone oxidoreductase core subunit S1 (Figure 6).

Links presently observed between microbiome-derived short-chain fatty acids and host somatic body mass assimilation are characterized in the literature and are primarily explained by bulk mitochondrial oxidative phosphorylation capacity enhancement in host skeletal muscle [50]. Transcriptome profiling revealed significantly enriched pathways in high muscle yield recolonized juveniles implicating electron transport chain activity and ATP production. Moreover, the uptake of short-chain fatty acids in liver and muscle cells is independent of fatty acid transport protein, plasma membrane fatty acid translocases, or cytosolic fatty acid binding proteins, as opposed to long or medium-chain fatty acids, which require energetically expensive shuttling into cells [51]. In germ-free mice, exogenous administration of butyrate resulted in a marked increase in the NADH/NAD+ ratio in colonocytes [14]. An increase in mitochondrial TCA activity should result in a greater fraction of reduced NADH, which functions as a substrate for the electron transport chain and downstream ATP production.

A third subset of enriched pathways in the high muscle yield recolonized larvae consisting of three highly enriched pathways for melanosome and pigment granule transport was also observed (Figure 6). Finally, upregulation of matrix metallopeptidase 9, integrin subunit beta 3, coagulation factor XIII, and adrenoceptor alpha 2b (Figure 6) revealed a group of enriched pathways implicated with wound healing and response to wound healing in the high muscle yield recolonized population (Figure 5). Upregulation of retinoic acid receptor, alpha b (*raraa*) was reported in low muscle yield recolonized larvae, a known DNA-binding transcription repressor (Figure 6). Additionally, nuclear receptor subfamily 1, group D, member 4b (*nr1d4b*) was found to be upregulated (Figure 6). Highly enriched pathways implicating positive regulation of protein containing complex disassembly receptor signaling pathway, and retinoic acid receptor signaling pathway were reported (Figure 5) concordant with *raraa* and *nr1d4b* variable transcription. A second category of pathway enrichment includes circadian rhythm (Figure 5). Additionally, a pathway for ER membrane and sub-compartment enrichment was noted. FoxO signaling pathway was significantly enriched in the low muscle yield recolonized group, as well as DNA binding transcription factor and transcription regulator activities (Figure 6).

Increased body mass accretion in the high muscle yield (ARS-FY-H) recolonized population is hypothesized to be mediated by commensal bacterial carbohydrate fermentation and short-chain fatty acids (SCRA) production. Additionally, certain gut flora profiles have been linked to enhanced nutrient absorption and weight gain resulting from villous vascularization I [52]. The microbiome has also been shown to regulate the triglyceride processing cascade by directly impacting fasting-induced adipose factor (*Fiaf*), resulting in the inhibition of lipoprotein lipase activity (*LPL*) and leading to insulin resistance and accumulation of triglycerides in adipose and hepatic tissues [53].

## 4. Conclusions

This study provides insight into intestinal colonization dynamics during the early life stages of rainbow trout, revealing muscle yield recolonized larvae to grow variably in mass depending on microbiome origin. Notably, high muscle yield recolonized trout displayed higher alpha diversity, and beta diversity clustering compared to low muscle yield recolonized trout, underpinning the complex interplay between the gut microbiome and muscle mass accretion. We reported links between respirasome, electron transport chain equivalent inputs, and mitochondrial upregulation with genetic line-specific microbiome in teleost fish. Future work may further cement the relationship between intestinal epithelia, and myocyte response to direct SCFA supplementation in vivo. Additionally, rainbow trout specific characterization of both microbial taxa and functional profile responsible for carbohydrate fermentation and SCFA production in cool, freshwater rearing environments may aid in future gut-engineering efforts, bolstering aquaculture sustainability through enhanced feed efficiency. A higher resolution snapshot of the interplay between host muscle and intestinal gut milieu may be possible using a high-quality reference metagenome, as well as extensive transcriptome and deep shotgun metagenome gut profiling approaches. Furthermore, the mechanisms underpinning host selection for unique microbial profiles remain opaque, and upcoming lines of investigation may reveal key insights linking host genotype and selective pressure impacting the seemingly heritable gut microbiome. Ultimately, the use of targeted prebiotics in broad-scale aquaculture production may supplement genetic selection for improved muscle yield, fast growth, and host resilience in the face of transient bacterial insult and increase economic yields through more efficient nutrient utilization in feeds.

## Figures and Tables

**Figure 1 microorganisms-12-00261-f001:**
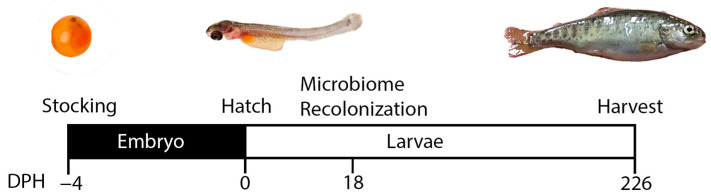
Fecal recolonization experiment timeline.

**Figure 2 microorganisms-12-00261-f002:**
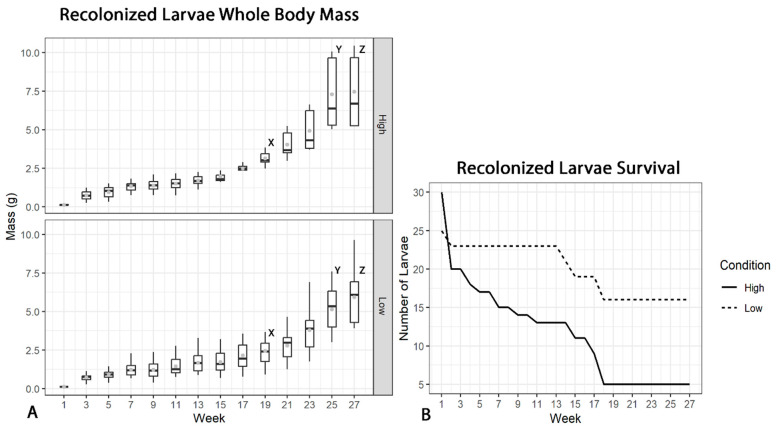
(**A**) High (ARS-FY-H) and Low (ARS-FY-L) muscle yield recolonized larvae average weight (g) over a 5-month growth period. Mean weight is displayed by grey points, median weight by horizontal lines. The high muscle yield recolonized group displayed significant pairwise body mass gain (*p* < 0.05) compared to low muscle yield larvae at 19, 25, and 27 weeks post-hatch denoted by X, Y, and Z, respectively. (**B**) Recolonized larvae survival data by group. Low muscle yield (ARS-FY-L) recolonized larvae displayed increased survival over 27 weeks.

**Figure 3 microorganisms-12-00261-f003:**
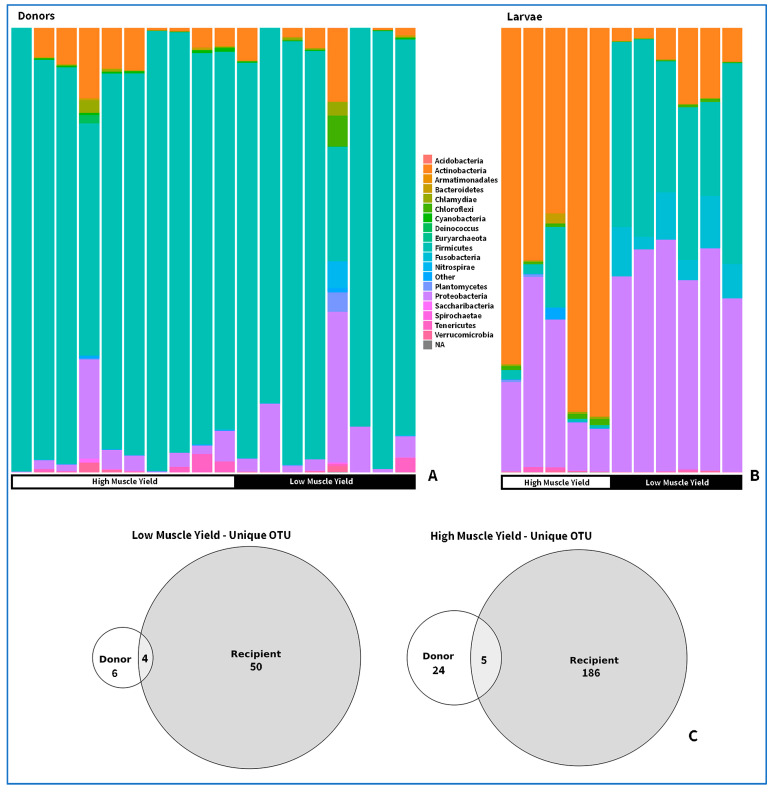
(**A**) Phylum composition of high/low muscle yield donors. (**B**) Phylum composition comparison between high/low muscle yield recolonized larvae. Phylum *Fusobacteria*, *Actinobacteria, Firmicutes*, *Proteobacteria*, and *Chloroflexi* were found to be variable abundant. (**C**) Number of shared OTUs between fecal donors and recipients.

**Figure 4 microorganisms-12-00261-f004:**
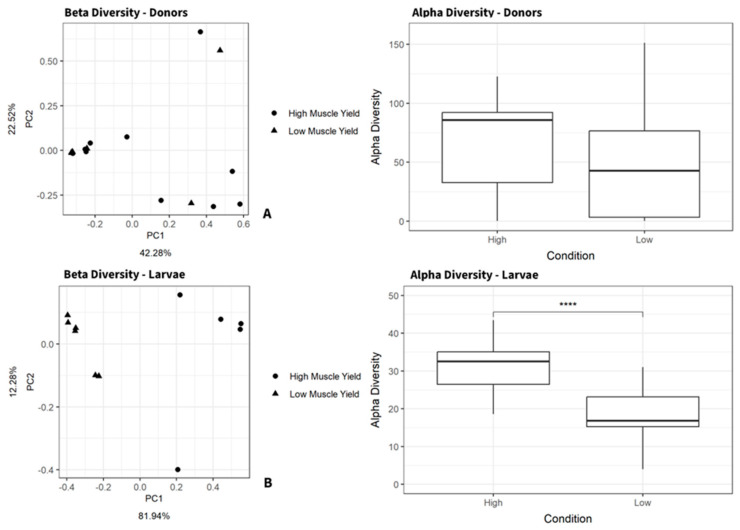
(**A**) Bray–Curtis beta diversity measures generated from adult donor fecal samples and larvae distal intestinal samples; clustering is observed for larvae fecal samples but not donor samples. (**B**) Fisher alpha diversity measures from adult donor fecal samples and larvae intestinal samples. Larvae alpha diversity was significant between conditions as denoted by the asterisks (Wilcoxon rank-sum pairwise test, *p* < 0.0001, two-sided) no significance was reported for fisher alpha diversity in donor samples.

**Figure 5 microorganisms-12-00261-f005:**
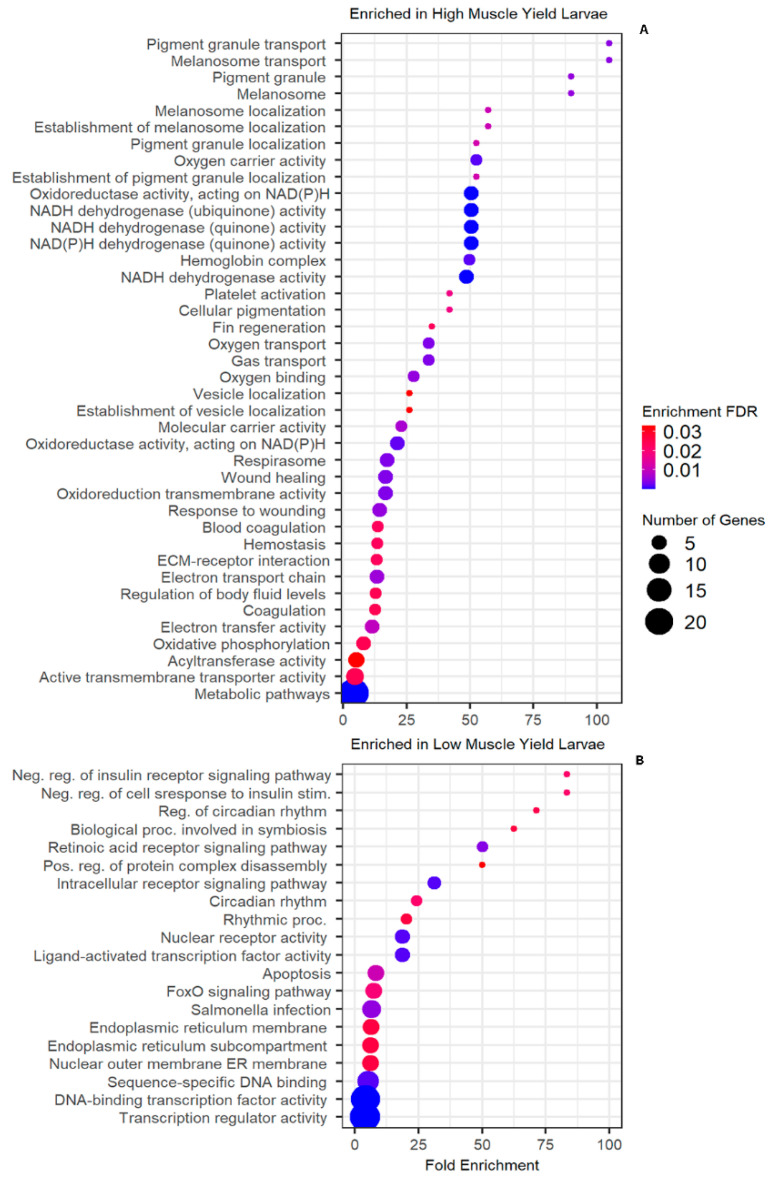
(**A**) Enriched GO terms found in high muscle yield recolonized larvae. Notable pathways include gas transport, NADH dehydrogenase activity, respirasome, and electron transport activity. (**B**) Enriched GO terms found in low muscle yield recolonized larvae. Significant pathways include negative regulation of the insulin receptor signaling pathway and retinoic acid receptor signaling pathway.

**Figure 6 microorganisms-12-00261-f006:**
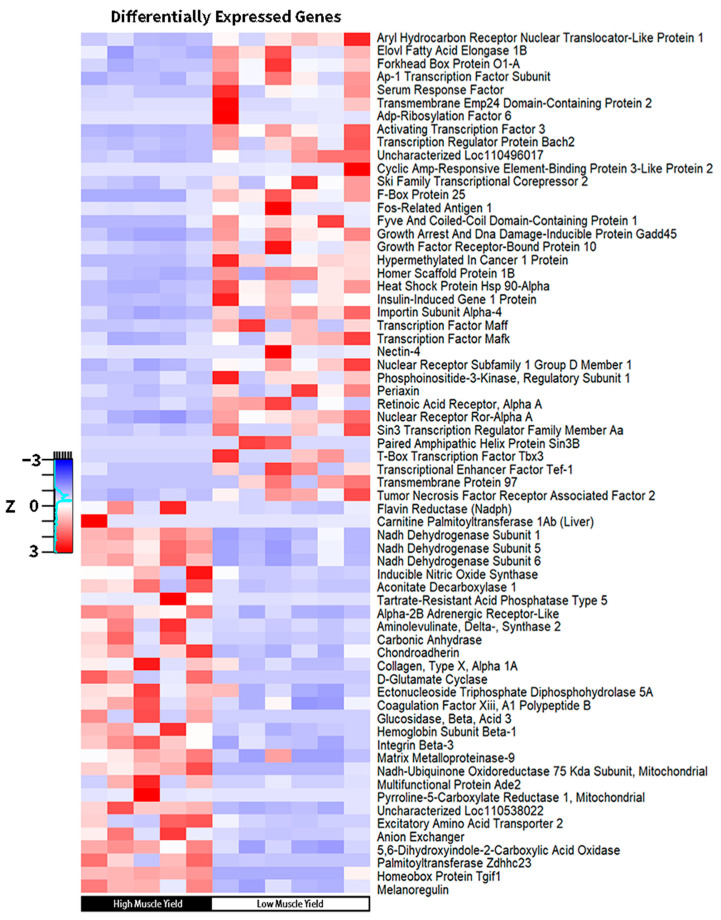
Heatmap for noteworthy differentially expressed genes involved in enriched pathways in high and low muscle yield recolonized larvae. Y axis denotes normalized Z scores generated using RPKM (reads per kilobase million) per sample.

## Data Availability

All data are provided in Appendix A. 16s rRNA amplicon data of rainbow trout intestines were submitted to the GenBank database in NCBI (National Center for Biotechnology Information) under the Bioproject ID PRJNA1034537. https://dataview.ncbi.nlm.nih.gov/object/PRJNA1034537?reviewer=1dmdkbdltn4s0rngm8avet3g6, accessed on 25 January 2024.

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
