# Peer review of "Early-Life Fecal Transplantation from High Muscle Yield Rainbow Trout to Low Muscle Yield Recipients Accelerates Somatic Growth through Respiratory and Mitochondrial Efficiency Modulation"

_microorganisms, 2024, doi:10.3390/microorganisms12020261_

Round 1

Reviewer 1 Report

Comments and Suggestions for Authors

This paper presents the findings from an experiment to investigate the effects of early-life fecal transplantation from high-muscle yield rainbow trout to low-muscle yield recipients and the authors hypothesized that the microbiome can accelerate somatic growth through respiratory and mitochondrial efficiency modulation. The authors present interesting results mainly associated with links between respirasome, electron transport chain equivalent inputs and mitochondrial upregulation with genetic line specific microbiome in teleost fish since the somatic mass was significantly increased in larvae receiving high fillet yield microbiome cocktails at 27 weeks post-hatch.  I believe there is much merit to the data presented in this paper. Therefore, I recommend its publication. However, a few problems were observed in the manuscript, and the following comments should be addressed:

- The title must not have a period (full stop)

- Line 125. Authors must format the font size “Waltham, MA”

- Lines 135 to 137 paragraph formatting must be corrected.

- Authors must point out in Figure 3 what 3A, 3B, and 3C are to facilitate readers' comprehension.

- Lines 192 to 200 describe the findings from the Wilcoxon rank sum test conducted on fisher alpha and beta diversity. Where is the Figure 4?

Author Response

Comments and Suggestions for Authors

This paper presents the findings from an experiment to investigate the effects of early-life fecal transplantation from high-muscle yield rainbow trout to low-muscle yield recipients and the authors hypothesized that the microbiome can accelerate somatic growth through respiratory and mitochondrial efficiency modulation. The authors present interesting results mainly associated with links between respirasome, electron transport chain equivalent inputs and mitochondrial upregulation with genetic line specific microbiome in teleost fish since the somatic mass was significantly increased in larvae receiving high fillet yield microbiome cocktails at 27 weeks post-hatch.  I believe there is much merit to the data presented in this paper. Therefore, I recommend its publication. However, a few problems were observed in the manuscript, and the following comments should be addressed:

- The title must not have a period (full stop)

We have amended the title to omit the period, the period has been added by the MDPI templet formatting

- Line 125. Authors must format the font size “Waltham, MA”

The text in line 125 has been amended to conform with the rest of the document.

- Lines 135 to 137 paragraph formatting must be corrected.

Paragraph formatting found between line 135 and 137 has been corrected to conform with the rest of the document.

- Authors must point out in Figure 3 what 3A, 3B, and 3C are to facilitate readers' comprehension.

Figure amended to include identifiers and more easily distinguish between 3a, 3b and 3c.

- Lines 192 to 200 describe the findings from the Wilcoxon rank sum test conducted on fisher alpha and beta diversity. Where is the Figure 4?

Figure 4 including fisher alpha diversity, as well as beta dispersion metrics for 16s data generated from recipients and donors have been added to the manuscript.

Reviewer 2 Report

Comments and Suggestions for Authors

Reviewer Comments for Manuscript titled Early-life fecal transplantation from high muscle yield rainbow trout to low muscle yield recipients accelerates somatic growth through respiratory and mitochondrial efficiency modulation

This study evaluated the effects of fecal microbiome from two genetic lines of rainbow trout on the intestinal microbiota of low-fillet-yield (ARS-FY-L) genetic lines in adult rainbow trout. It is an interesting study that utilizes the concept of fecal microbiota transplantation to investigate the function of gut microbiota. My review comments are as follows:

1. The title is too long. From an M&M, this may not be considered as fecal transplantation but rather than oral fecal microbiome, as feed enters the intestine through the stomach and the core microbial community may undergo changes through digestion.

2. Introduction, line33-35, I do not understand what do you mean by saying this sentence. What are unique populations? What do you mean by “geographically isolated”?

Line 63, I don’t think it can be defined as “microbiome-depleted rainbow trout”.

3. Line 76-78, I do not understand what the authors mean by saying this sentence. Rainbow trout muscle yield genetic lines are shown to be predictive of distinct gut microbial assemblages? How can it predict?

4. The expressions of some sentences are obscure and difficult to understand. I strongly recommend using concise language to express a meaning correctly, so that non-native English speakers can also understand it.

For instance, line 56-57, “Despite evidence in mammalian models, the direct role of gut microbiome 56 composition on energy regulation and skeletal muscle recruitment remains opaque in agriculturally relevant animals, including rainbow trout” , what are agriculturally relevant animals? Does it not include mammals? In addition, I have to check the meaning of the word "opaque" by using translation software.

 5. Methodology

The difference in growth stage and environmental conditions may influence the composition of gut microbiota. What is the living condition of adult donors. How did you treat the fecal microbiome samples? how did you get them? What is the proportion of these fecal samples in the feed? Has it been tested for pathogens? Gut microbiota may change with growth stage, how did you sample them and store them?

Why do you sample the whole tail tissue for RNA-seq analysis?  Is there a correlation between fish tail traits and gut microbiota composition?

How did you calculate and compare muscle mass? and body weight? You should at least mention it in the method.

6. Results & Discussion

In Line 148, it is suggested that the authors should add the analysis the specific growth rate of all the groups of Rainbow trout.

From Figure2 B, Rainbow trout have a higher survival rate when consuming feces from the same genetic line? The survival rates of rainbow trout in both treatment groups were very low. Do you have a control group fed with non-fecal feed?

Line 335-350, the differentially expressed genes used for the enriched pathways, and it is suggested that the authors should choose several genes to verity the results of RNA Sequencing.

Comments on the Quality of English Language

Author Response

Comments and Suggestions for Authors

Reviewer Comments for Manuscript titled Early-life fecal transplantation from high muscle yield rainbow trout to low muscle yield recipients accelerates somatic growth through respiratory and mitochondrial efficiency modulation

This study evaluated the effects of fecal microbiome from two genetic lines of rainbow trout on the intestinal microbiota of low-fillet-yield (ARS-FY-L) genetic lines in adult rainbow trout. It is an interesting study that utilizes the concept of fecal microbiota transplantation to investigate the function of gut microbiota. My review comments are as follows:

  1. The title is too long. From an M&M, this may not be considered as fecal transplantation but rather than oral fecal microbiome, as feed enters the intestine through the stomach and the core microbial community may undergo changes through digestion.

We believe the title of fecal transplantation is warranted for this study as oral administration of exogenous fecal material is commonly described using these terms in the literature. For instance, in the Chenyan study cited below, Zebrafish were fed fecal microbiomes originating from other fish. Also, in the Rawls study cited here, germ-free mice were recolonized with native zebrafish microbiome following a gavage feeding strategy.  Note, both studies describe oral fecal microbiome assimilation as “fecal transplantation”.

Chenyan Hu, Baili Sun, Mengyuan Liu, Junxia Yu, Xiangzhen Zhou, Lianguo Chen, Fecal transplantation from young zebrafish donors efficiently ameliorates the lipid metabolism disorder of aged recipients exposed to perfluorobutanesulfonate, Science of The Total Environment, Volume 823, 2022, https://doi.org/10.1016/j.scitotenv.2022.153758.

Rawls JF, Mahowald MA, Ley RE, Gordon JI. Reciprocal gut microbiota transplants from zebrafish and mice to germ-free recipients reveal host habitat selection. Cell. 2006 Oct 20;127(2):423-33. doi: 10.1016/j.cell.2006.08.043. PMID: 17055441; PMCID: PMC4839475.

  1. Introduction, line33-35, I do not understand what do you mean by saying this sentence. What are unique populations? What do you mean by “geographically isolated”?

Geographically isolated populations describe groups of fish separated by large distances, meaning there is no direct contact or possibility of microbiome transfer between the populations.

Line 63, I don’t think it can be defined as “microbiome-depleted rainbow trout”.

We reared rainbow trout eggs following a gnobiotic protocol described by Perez-Pascal 2021 wherein the author described the microbiome depleted rainbow trout rearing protocol used herein. These animals are raised in sterilized, autoclaved water laced with antibiotics.

Pérez-Pascual, D., et al., Gnotobiotic rainbow trout (Oncorhynchus mykiss) model reveals endogenous bacteria that protect against Flavobacterium columnare infection. PLOS Pathogens, 2021. 17(1): p. e1009302.

  1. Line 76-78, I do not understand what the authors mean by saying this sentence. Rainbow trout muscle yield genetic lines are shown to be predictive of distinct gut microbial assemblages? How can it predict?

Previous work conducted by our lab found rainbow trout genetic line to be predictive of microbial gut profile. Specifically high muscle yield genetic lines contained greater firmicutes content. Nonetheless we have rephrased this statement to avoid confusion to read “Non-invasive microbial gut sampling may assist in future precision breeding efforts as rainbow trout muscle yield genetic lines are shown to harbor distinct gut microbial assemblages”.

Chapagain, P., et al., Distinct microbial assemblages associated with genetic selection for high- and low- muscle yield in rainbow trout. BMC Genomics, 2020. 21(1): p. 820.

  1. The expressions of some sentences are obscure and difficult to understand. I strongly recommend using concise language to express a meaning correctly, so that non-native English speakers can also understand it.

For instance, line 56-57, “Despite evidence in mammalian models, the direct role of gut microbiome 56 composition on energy regulation and skeletal muscle recruitment remains opaque in agriculturally relevant animals, including rainbow trout” , what are agriculturally relevant animals? Does it not include mammals? In addition, I have to check the meaning of the word "opaque" by using translation software.

We have removed terms that may cause confusion to non-native readers, specifically, removing the term opaque from line 56-57. The sentence now reads “Despite work in murine model species, the direct role of gut microbiome composition on energy regulation and skeletal muscle recruitment remains unclear in agriculturally relevant animals, including rainbow trout”

  1. Methodology

The difference in growth stage and environmental conditions may influence the composition of gut microbiota. What is the living condition of adult donors. How did you treat the fecal microbiome samples? how did you get them? What is the proportion of these fecal samples in the feed? Has it been tested for pathogens? Gut microbiota may change with growth stage, how did you sample them and store them?

We added these details as follows “Fecal microbiome samples collected from apparently healthy high (ARS-FY-H)  and low (ARS-FY-L)  muscle yield donors at approximately 26 months of age housed in a common recirculating aquaculture system (RAS) were mixed with feed in a 2 mL tube after 15 seconds of vortexing.  Feed laden with active microbiome was provided to larvae at first feeding and daily for 27 weeks. Fecal samples were collected from adult donors weekly by manually stripping adult donors. Fecal samples were stored in a 15 mL PBS solution at 4 oC and 3mL were incorporated with recipient feed daily. Raw sub-samples not kept in PBS were kept at -80 oC for microbiome analysis. Juvenile fish were fed 15% body weight daily throughout the experiment, and biometrics were recorded weekly by generating individual wet-weight measurements”. 

Why do you sample the whole tail tissue for RNA-seq analysis?  Is there a correlation between fish tail traits and gut microbiota composition?

Whole tail samples were generated to interrogate the muscle transcriptome. We were interested in finding underlying biomarkers that may explain the difference in body mass observed between the two groups. Fish were too small to dissect individual organs.

How did you calculate and compare muscle mass? and body weight? You should at least mention it in the method.

We did not compare muscle mass, just whole body weight “Juvenile fish were fed 15% body weight daily throughout the experiment, and biometrics were recorded weekly by generating individual wet-weight measurements.  At 27 weeks post-hatch, fish were euthanized, and whole tail sections from 11 juvenile fish were used for RNA-seq analysis”.

  1. Results & Discussion

In Line 148, it is suggested that the authors should add the analysis the specific growth rate of all the groups of Rainbow trout.

Specific growth rate data added “The standard growth rate (SGR) was 1.47% of final biomass in the high, and 1.37% in the low groups respectively”. 

From Figure2 B, Rainbow trout have a higher survival rate when consuming feces from the same genetic line? The survival rates of rainbow trout in both treatment groups were very low. Do you have a control group fed with non-fecal feed?

Actually, the survival rate was higher for larvae receiving fecal feces from the same genetic line (high donor line to high larvae line). The survival rate was lower when feces transferred from high line to low line. Survival rate was likely low due to the early life gnobiotic treatment especially with mismatching genetics.  No control group was used for the present study, we recognize this to be a limitation, and aim to incorporate a control condition in follow up studies. We discussed the survival rate in the text

“The 3-fold increase in mortality observed in the high muscle yield recolonized larvae group may be explained by the abundance of Botulinum spp in this population, which has been linked to mortality in a wide range of freshwater teleost species [32]. Moreover, botulinum spp. Was not detected in the low muscle yield (ARS-FY-L) group. Mortality in the high (ARS-FY-H) muscle yield recolonized group was most pronounced during the first 13 weeks of the experiment.  Early life stage rainbow trout are known to be more susceptible to bacterial infections, with limited immune response ability via dynamic gene regulation and transcription modulation compared to mature conspecifics [33].  For instance, enteric red mouth disease (ERM) induced by Yersinia ruckeri has been demonstrated to be highly lethal in juvenile rainbow trout despite susceptibility across life stages [34]”. 

Line 335-350, the differentially expressed genes used for the enriched pathways, and it is suggested that the authors should choose several genes to verity the results of RNA Sequencing.

We believe our RNA-seq results to be robust and not require qPCR validation. Our lab has published 20+ studies including RNA-seq results starting in 2015 and we have always found strong concordance between RNA-seq and qPCR results. The RNA-Seq technique has matured enough that it does not require further validation unless interesting genes are warranted additional analysis.